# Morphometric Traits as Predictors of Body Mass in the Marine Gastropod *Semicassis bisulcatum*: Insights for Aquaculture and Selective Breeding

**DOI:** 10.3390/ani15203027

**Published:** 2025-10-18

**Authors:** Dewei Cheng, Yun Chen, Xin Liu, Shiyuan Bao, Xuyang Chen, Ying Qiao, Ersha Dang

**Affiliations:** 1Key Laboratory of Tropical Marine Ecosystem and Bioresource, Fourth Institute of Oceanography, Ministry of Natural Resources, Beihai 536015, China; chengdewei@4io.org.cn (D.C.); liuxin@4io.org.cn (X.L.); baoshiyuan@4io.org.cn (S.B.); chenxuyang@4io.org.cn (X.C.); qiaoying@4io.org.cn (Y.Q.); 2Guangxi Key Laboratory of Beibu Gulf Marine Resources, Environment and Sustainable Development, Fourth Institute of Oceanography, Ministry of Natural Resources, Beihai 536015, China; 3Ecological Environment Monitoring and Scientific Research Center, Taihu Basin & East China Sea Ecological Environment Supervision and Administration Bureau, Ministry of Ecology and Environment, Shanghai 200125, China; chenyun@thdhjg.mee.gov.cn

**Keywords:** semicassis bisulcatum, morphological traits, mass traits, correlation analysis, path analysis, multiple regression analysis, selective breeding

## Abstract

The gastropod *Semicassis bisulcatum* represents a promising candidate for aquaculture. However, essential biological data for establishing selective breeding programs are still lacking. This study quantified morphological and mass traits in wild populations and revealed that mass-related traits, with soft body mass being the most notable, show substantially greater individual variation than morphological traits. This finding indicates significant potential for genetic improvement. Shell thickness was identified as the strongest predictor of total body mass, while shell width was most closely associated with soft tissue yield. These relationships provide concrete breeding guidance, suggesting that selecting for thicker shells enhances overall biomass, whereas prioritizing wider shells improves soft tissue production. This work establishes the first phenotypic framework for selective breeding in this species, supporting its sustainable aquaculture development.

## 1. Introduction

*Semicassis bisulcatum* (Schubert & Wagner, 1829), a marine gastropod belonging to the family Cassidae, is characterized by its distinct globose shell with prominent spiral grooves and ridges. This species exhibits a broad distribution throughout the Indo-West Pacific region, including coastal waters of China, Taiwan, Japan, and Vietnam [1]. It predominantly occurs on sandy or muddy substrates at depths ranging from the subtidal zone to approximately 100 m [2]. As a carnivorous species known to prey on echinoderms, *S. bisulcatum* confers a dual economic potential. Firstly, as a species occupying a higher trophic level, its soft tissue is typically rich in protein, suggesting its potential as a high-value aquatic product. More significantly, this specific predatory behavior enables its utilization as a natural biocontrol agent within integrated aquaculture systems. For instance, it could be employed to manage populations of detrimental sea urchins in sea cucumber or abalone cultivation ponds, thereby enhancing the yield and economic returns of the primary cultured species through ecological mechanisms [3]. However, in recent years, despite its underdeveloped potential for aquaculture, wild populations of *S. bisulcatum* have faced increasing pressures from overharvesting for commercial collection as ornamental shells and seafood products, coupled with habitat degradation. While a lack of fundamental biological studies and aquaculture technologies has led to a consistent decline in population numbers [4].

In shellfish research, morphological traits and mass traits are widely recognized as critical indicators of growth performance and are frequently used as primary targets in selective breeding programs [5,6]. The establishment of quantitative relationships among these traits through methods such as correlation analysis, path analysis, and regression modeling not only facilitates the identification of key morphological proxies for biomass prediction but also provides deeper insights into growth and developmental patterns [7,8,9,10]. This approach has been successfully applied to guide broodstock selection and breeding strategy design in several economically important mollusks, including *Pomacea canaliculata* [11], *Bellamya purificata* [12], *Volutharpa ampullaceal* [13], *Neverita didyma* [14], *Bellamya aeruginosa* [15]. In these species, the relationships between morphological and mass traits have been systematically quantified, establishing a solid foundation for selective breeding and aquaculture optimization [16,17]. In contrast, the research on *S. bisulcatum* remains confined to preliminary taxonomic descriptions and basic habitat records, with a notable scarcity of fundamental biological data, particularly concerning its growth patterns. As understanding growth patterns is crucial for elucidating life history strategies and formulating aquaculture protocols, this knowledge gap significantly hinders the assessment of its aquaculture potential and production efficiency [18,19].

Therefore, prior to establishing commercial aquaculture for this species, it is essential to characterize its fundamental growth patterns and identify key morphological indicators that can reliably predict biomass traits, including body mass and soft tissue mass. The application of morphometric and correlation analyses to quantify trait relationships represents the most direct and feasible research approach to achieve this goal.

To address this research gap, the present study conducted a systematic examination of wild *S. bisulcatum* populations from the Beibu Gulf region. Specifically, the seven morphological traits and two mass traits have been measured to analyze the interrelationships among key morphological and mass traits and identify morphological features with the greatest direct contribution to body mass and soft body mass, and predictive regression models have been constructed for enabling biomass based on morphometric data. The findings of this study could provide valuable insights into the phenotypic determinants of growth in *S. bisulcatum*, thereby supporting future initiatives in resource conservation, aquaculture development, and genetic breeding programs of this species.

## 2. Materials and Methods

### 2.1. Sample Collecting

A total of 150 wild *S. bisulcatum* individuals were collected in July 2024 (during the species’ rapid growth season) from the coastal waters around Weizhou Island in the Beibu Gulf, China (sampling area coordinates: 109.1682–109.2713° E, 20.8716–20.9217° N) (Figure 1). Sampling was performed via SCUBA diving within the depth zone of 6.5 to 20 m. From the collected individuals, 100 adult individuals with intact shells, no visible damage, and vigorous activity were selected as experimental subjects to minimize bias resulting from injury or developmental stage variation. Live specimens were transported to the laboratory within 2 h in oxygenated seawater and acclimatized for 3 days in a recirculating aquaculture system (temperature: 30 °C, salinity: 30‰). No feeding was provided during the acclimation period. All morphometric and mass measurements were initiated following the completion of the three-day acclimation phase.

The water quality parameters recorded at the sampling area were as follows: water temperature 30.12 ± 0.21 °C, water depth (6.5–20 m) ± 0.54 m, salinity 33.3 ± 0.15‰, pH 8.14 ± 0.24, dissolved oxygen 6.53 ± 0.15 mg/L, and suspended solids 6.7 ± 0.72 mg/L.

### 2.2. Measurement of Morphological Traits

A total of 100 healthy and vigorously active individuals of *S. bisulcatum* were randomly selected as experimental subjects. All specimens were processed in a fresh state immediately after the acclimation period. Prior to morphological measurements, each specimen was blotted dry with absorbent paper to remove surface moisture for accurate body mass determination. Total body mass (including shell) was measured using an electronic balance (Hochoice: Shanghai Hochice Industrial Co., Ltd., Shanghai, China) with an accuracy of 0.001 g.

The seven morphometric traits (Figure 2) including shell height (SH), body whorl height (BWH), spire height (SPH), aperture height (AH), aperture width (AW), shell width (SW), and shell thickness (ST) were measured (detail in Table 1) using IP67 digital display Vernier calipers (Deli: Ningbo Deli Group Co., Ltd., Ningbo, China), accurate to 0.01 mm.

For soft body mass determination (including gonads and visceral mass), specimens were carefully dissected using stainless steel surgical tools to separate the complete soft tissue from the shell. The soft tissue was then blotted on absorbent paper to remove excess moisture without desiccation, and immediately weighed using the same electronic balance (accuracy: 0.001 g). All measurements were completed within 30 min of dissection to prevent tissue dehydration affecting mass accuracy.

### 2.3. Statistical Analysis

Descriptive statistics for all morphological and mass traits, including the mean, standard deviation (SD), coefficient of variation (CV), skewness, and kurtosis, were conducted using SPSS (Version 21.0). the coefficient of variation (CV) was computed as follows: CV = (SD/mean) × 100%. according to GOMES (1985) [20], the CV is classified as low (CV < 10%), medium (CV between 10% and 20%), high (CV% between 20% and 30%), and very high (CV > 30%).

To validate the assumptions of subsequent parametric analyses, normality of the dependent variables (body mass and soft body mass) was assessed using the Kolmogorov–Smirnov (K–S) test [21]. Subsequently, Pearson correlation analysis was applied to examine bivariate relationships among all traits. To control the inflated family-wise error rate resulting from multiple comparisons (36 pairwise correlation tests in total), we applied the conservative Bonferroni correction. This adjustment yielded a revised significance threshold of *p* < 0.00139 (α = 0.05/36).

To further elucidate the causal structure underlying these relationships, path analysis was performed according to established methodological frameworks [22]. This approach decomposes correlation coefficients into direct and indirect effects. The relative contribution of each morphological trait was quantified using determination coefficients, representing the proportion of variance in mass traits explained through direct and indirect pathways. These coefficients were calculated as follows:$$d_{i}=P_{i}^{2}$$$$d_{ij}=2r_{ij}P_{i}P_{j}$$
where *d_i_* is the direct determination of *i*th trait on the body mass and *d_ij_* is the codetermination of *i*th trait on the body mass through *j*th trait (*i* ≠ *j*); *P_i_* and *P_j_* are the path coefficients of *i*th and *j*th trait on the body mass, respectively; and *r_ij_* is the correlation coefficient between *i*th and *j*th trait.

A backward stepwise multiple regression approach was implemented, using a significance threshold of α = 0.05 for variable removal, to identify the most parsimonious combination of morphological traits for predicting each mass trait. The procedure initialized with all candidate predictors in the model and iteratively eliminated the least significant contributor (determined by the highest *p*-value) at each iteration, terminating when all retained variables reached statistical significance (*p* < 0.05). Finally, curve estimation regression was implemented between the selected morphological predictors and mass traits to determine the optimal functional form of these relationships [23].

## 3. Results

### 3.1. Descriptive Statistics of Parameters and Correlation Coefficients

The mean, SD, and CV for the morphometric traits and mass traits of wild *S. bisulcatum* are presented in Table 2. the mass traits exhibited significantly higher coefficients of variation (CV) than morphological traits, indicating greater variability among individuals. The soft body mass showed the highest CV (41.04%), followed by body mass (23.88%). Among morphological traits, SH had the highest CV (21.78%), while ST was the most stable (6.90%).

Notably, dissection-based assessment of reproductive status revealed that 70% of individuals possessed developed gonads, suggesting our July sampling period coincided with the peak reproductive season. This reproductive activity likely contributes to the high variability observed in soft body mass, as energy allocation to gonad development introduces additional variation beyond somatic growth alone. These results collectively indicate that mass traits, particularly soft body mass, hold substantial breeding potential and should be prioritized as key targets in selective breeding programs, though seasonal reproductive cycles should be accounted for in future sampling designs.

### 3.2. Correlation Analysis Between Various Traits

The vast majority of correlation coefficients among the traits in wild *S. bisulcatum* reached a highly significant level (*p* < 0.05), indicating the statistical relevance of the selected traits for correlation analysis. The correlation coefficients of various traits were shown in Figure 3. The result showed that the correlation coefficients between the morphological traits and body mass were ranked as follows: ST = SW > AH > SH> BWH > AW > SPH. Among these traits, SH, AH, SW, BWH, AW and ST showed strong correlations (r > 0.7) with body mass, and SPH were moderately correlated (0.4 < r < 0.7).

Similarly, the correlation coefficients with soft body mass decreased in the order: SW > ST > AH > SH > BWH > AW > SPH. Again, strong correlations (r > 0.7) were observed for SH, AH, SW, and ST, moderate correlations (0.4 < r < 0.7) were found for BWH, SPH and AW.

These results indicated that shell height, aperture height, shell width, and shell thickness are the key morphological traits that are strongly associated with both body mass and soft body mass in *S. bisulcatum*. In contrast, spire height showed minimal influence. This finding provides critical trait references for the selective breeding of this species.

### 3.3. Regression Analysis Between Morphological Traits and Mass Traits

A backward stepwise regression analysis was conducted to identify morphological traits with significant effects on the dependent variables Y_1_ and Y_2_. The path coefficients for these significant independent variables were subsequently estimated (Table 3). For body mass, the analysis revealed that ST, AH, and SH were retained in the model due to their significant effects (*p* < 0.05), while BWH, SPH, AW, and SW were excluded because of lacking statistical significance. For soft body mass, SW and AH were retained as significant predictors (*p* < 0.05), whereas SH, BWH, SPH, AW, and ST were removed. The resulting optimal regression models for *S. bisulcatum* (see Table 3 for coefficients) are as follows:Y_1_ = −43.543 + 1.242 ST + 0.398 AH + 0.339 SH (*R*^2^ = 0.853)Y_2_ = −30.859 + 0.696 SW + 0.545 AH (*R*^2^ = 0.675)

In these equations, *R*^2^ represents the coefficient of determination, which quantifies the proportion of variance in the dependent variable explained collectively by the independent variables. A value closer to 1 indicates a better model fit [10]. The *R*^2^ value of 0.853 indicates that 85.3% of the variation is accounted for by ST, AH, and SH in body mass. Similarly, the *R*^2^ of 0.675 indicates that SW and AH explain 67.5% of the variability in soft body mass.

To further validate that the selected morphological traits represent the major factors influencing mass traits, analysis of variance was performed on the regression models (Table 4). The results confirmed that ST, AH, and SH were key morphological determinants of body mass, while SW and AH were predominant factors affecting soft body mass. These findings demonstrated that the derived regression equations effectively captured the relationship between morphological traits and mass traits, providing a robust quantitative basis for the selective breeding of *S. bisulcatum*.

### 3.4. Path Analysis of Morphological Traits on Mass Traits

Path coefficient reflects the strength of the direct associations between independent variables and dependent variables. As shown in Table 5, ST exhibited the strongest direct effect on body mass, with a path coefficient of 0.509, followed by SH at 0.263, while AH showed the weakest direct effect at 0.220. For soft body mass, SW had the highest direct influence (path coefficient = 0.482), while AH had a relatively smaller direct effect (0.394). These results suggested that in breeding practices for *S. bisulcatum*, ST should be prioritized as a selection criterion for increased body mass, whereas SW is recommended as the primary screening trait for improving soft body mass, as these traits show the strongest statistical associations with the respective mass traits.

Analysis of indirect effects revealed that SH had the strongest indirect influence on body mass (0.595), followed by AH (0.582), while ST showed the smallest indirect contribution (0.389). For soft body mass, AH demonstrated the greatest indirect effect (0.365), whereas SW had a relatively lower indirect influence (0.298). These findings indicated that SH had a synergistic selective effect on body mass, while AH played a similar role in soft body mass.

Based on the path analysis, a breeding strategy combining primary and secondary traits is recommended. In body mass selection, ST should be served as the core indicator, with simultaneous consideration of SH. For soft body mass improvement, SW should be the main selection criterion, supplemented by attention to AH. This integrated selection approach can significantly enhance the efficiency and effectiveness of breeding programs for *S. bisulcatum*.

### 3.5. Determinative Influence of Morphological Traits on Mass Traits

The total direct determinative coefficient of the morphological traits on mass traits in *S. bisulcatum* was 0.857, while the total indirect determinative coefficient was 0.672 (Table 6). This result was consistent with the coefficient of determination (*R*^2^ = 0.857) derived from the multiple regression equation, further confirming that the selected traits—SH, AH, SW, and ST—collectively represent the major factors influencing body bass in this species.

Analysis of the direct determinative coefficients for individual traits revealed that ST had the strongest direct effect on body mass, with a coefficient of 0.259, followed by SH at 0.067, while AH exhibited the smallest direct contribution (0.048). For soft body mass, SW showed the highest direct determinative coefficient (0.232), while AH had a relatively lower value (0.154).

Regarding indirect effects, the combined determinative coefficient of SH and ST on body mass was the highest (0.313), followed by that of SH and AH (0.115). The combination of AH and ST had the smallest joint contribution (0.073). Additionally, AH and SW also showed a considerable combined effect on soft body mass, with a determinative coefficient of 0.286.

These results demonstrate that SH and ST were the principal traits determining body mass in *S. bisulcatum*, while AH and SW were the key morphological traits influencing soft body mass. This finding further validated the reliability of the path analysis conducted above.

### 3.6. Curve Regression Modeling

Curve fitting was performed using SPSS 22.0. The significant predictors—SH, AH, ST, and SW—previously identified from two multiple regression models, were employed as independent variables. Separate curve models were fitted with Y1 and Y2 as the dependent variables. As shown in Figure 4 and Figure 5, the optimal fit for the relationships between predictors (SH, AH, ST, SW) and each response variable (body mass, soft body mass) was achieved using cubic polynomial models.

## 4. Discussion

### 4.1. Analysis of Breeding Potential for Mass Traits in S. bisulcatum

This study systematically evaluated the relationships between morphological and body mass traits in the marine *S. bisulcatum*. The results indicate that, mass traits—particularly body mass and soft body mass exhibited considerable potential for selective breeding. The coefficient of variation (CV) for soft body mass (41.04%) was markedly higher than that for total body mass (23.88%), with both substantially exceeding the variability observed in all measured morphological traits (CV < 22%). These findings align with reports in other economically important mollusks, such as *Babylonia areolata* [24], *Thais clavigera* [25], and *Cipangopulusina chinensus* [26], reinforcing the view that mass traits generally demonstrate high phenotypic plasticity in gastropods.

The coefficient of variation serves as a key indicator of phenotypic dispersion. High CV values suggest broad variability within the population, which may be subject to strong genetic control or environmental regulation [27]. The notably high variability in soft body mass could reflect heightened sensitivity to environmental factors such as feed quality and water temperature fluctuations, or may indicate underlying genetic diversity [28]. Body mass, as a comprehensive measure of overall growth performance, also showed significantly greater variability than morphometric traits, implying substantial potential for enhancing growth rate and yield through genetic improvement [29].

Compared to the aforementioned species, *S. bisulcatum* demonstrates comparable or greater variability in mass traits, underscoring its promise as a candidate for selective breeding. These results provide a theoretical foundation for future family-based or population-based breeding programs [30]. It is recommended that selection prioritize soft body mass and body mass, while also integrating key morphological traits such as shell width and thickness to improve overall economic value. Furthermore, the rich phenotypic variation observed supports subsequent genetic analyses, including quantitative trait locus (QTL) mapping or genome-wide association studies (GWAS) [31].

### 4.2. Relationships Between Key Morphological and Mass Traits in S. bisulcatum and Their Biological Implications

Building on the statistical relationships established through backward stepwise regression, we identified key morphological traits that are predictive of mass traits in *S. bisulcatum* and interpret these within an integrated ecological and physiological framework. Specifically, shell height (SH), aperture height (AH), and shell thickness (ST) were highly significant predictors of total body mass (*p* < 0.01), while shell width (SW) and AH predominantly influenced soft body mass. These distinct association patterns suggest trait-specific morphological integrations that reflect fundamental adaptive strategies [5].

Ecological adaptation and morphological function constitute primary determinants. As a benthic carnivore inhabiting sandy-muddy substrates, the substantial shell (with ST being a key predictor of body mass) not only directly contributes to biomass but likely represents an adaptive strategy against predation and substrate interaction [32]. This structural investment necessarily directs specific energy allocation patterns [33]. Unlike congeneric species such as *Babylonia areolata*, SW was not retained in the body mass prediction model, indicating that weight accumulation in this species relies more heavily on vertical shell development. This structural specialization towards vertical development, as opposed to the lateral expansion observed in other gastropods, underscores how niche-specific selective pressures can shape distinct morphological solutions to common challenges of protection and resource allocation [34].

Energy allocation trade-offs profoundly influence trait expression. The high variability in soft body mass correlates with its specialized predation on echinoderms. This trophic specialization may lead to unpredictable food resource availability, consequently generating substantial inter-individual variation in energy reserves. The coordinated action of AH and SW in defining visceral capacity creates a physical framework that both constrains and facilitates energy storage—a crucial adaptation for a carnivore facing fluctuating prey availability [35]. This morphological integration effectively translates feeding success into growth potential within defined spatial parameters.

Reproductive investment represents another crucial regulatory variable. The observed high proportion of individuals (70%) with developed gonads aligns with reproductive patterns in cassid species. The development and maintenance of reproductive tissues require substantial energy investment, potentially creating a trade-off with somatic growth in soft tissues. Although ST closely associates with total body mass, its non-significant contribution to soft body mass suggests that shell calcification and soft tissue production may follow distinct regulatory pathways [36]. This apparent dissociation between structural and soft tissue investment has direct implications for selective breeding: it suggests that shell and tissue traits could be improved relatively independently, provided such trade-offs are managed within a balanced breeding objective.

These findings provide integrated guidance for breeding practices: selection for overall yield should emphasize SH, AH, and ST, while improvement of soft tissue yield should focus on SW and AH. We recommend developing a multiple-trait selection index to balance potential antagonisms among traits and maximize genetic gain [37]. Compared to congeneric species, the specific morphometric patterns exhibited by *S. bisulcatum* reflect its unique adaptation to particular ecological niches. Future cross-species comparative studies will better elucidate the evolutionary forces driving the diversification of these growth patterns.

### 4.3. Interpreting the Associations Between Morphological Traits and Mass Traits

Path analysis further elucidated the direct and indirect pathways of association between morphological traits and mass traits. Building on these statistical relationships, we interpret these pathways within an integrated framework of structural constraints, ecological adaptation, and energy allocation strategies.

For body mass, shell thickness (ST) exhibited the strongest direct association, which is consistent with its role in contributing to biomass through enhanced structural integrity [38]. This finding carries clear ecological significance: as a benthic species inhabiting sandy-muddy substrates, the thick shell likely represents a key adaptive trait against predation and environmental abrasion, with this structural investment directly translating into biomass accumulation. Although shell height (SH) showed a moderate direct effect, it demonstrated substantial indirect influence, suggesting that SH not only contributes directly to biomass but may also promote growth through synergistic relationships with correlated traits such as aperture height (AH) [39]. Aperture height (AH) operated primarily through indirect pathways, potentially regulating energy acquisition by influencing physiological processes like feeding efficiency or locomotion [35]. Collectively, these association pathways illustrate how structural defense (ST), spatial development (SH), and functional efficiency (AH) are interrelated aspects of growth that correlate with overall biomass accumulation.

Complementing the patterns observed for total mass, shell width (SW) demonstrated the strongest direct effect on soft body mass, implying that a wider shell directly provides greater accommodation space for soft tissue growth [39]. This result is directly relevant to energy allocation strategies: in an ecological context of unpredictable feeding opportunities, the wider visceral cavity provides the physical foundation for energy storage (e.g., developed digestive glands and gonads), which may represent an important adaptation for individuals occupying a carnivorous ecological niche. Aperture height (AH) functioned mainly through indirect mechanisms, possibly by enhancing feeding capacity or metabolic efficiency [35].

Decision coefficient analysis confirmed that these morphological traits collectively accounted for a substantial proportion of variance in both mass traits. Beyond confirming the predictive utility of these traits, our analysis provides mechanistic insights into how distinct shell architectures support different aspects of biomass allocation in *S. bisulcatum*. These outcomes partially diverge from previous studies, possibly reflecting the integration of unique shell formation mechanisms and specific ecological adaptation strategies—where the thick shell (ST) serves both structural support and direct biomass composition, while the wider shell form (SW) provides optimized space for energy storage and reproductive investment.

These findings emphasize the necessity for species-specific breeding strategies: selection for overall yield should prioritize shell thickness, while improvement of soft tissue yield should focus on shell width, with simultaneous consideration of synergistic effects involving shell height and aperture height. Deeper understanding of the biological mechanisms underlying target traits will facilitate the design of more efficient multiple-trait selection programs.

### 4.4. Limitations and Future Research Directions

This study has several limitations. First, as all samples were derived from wild populations, environmental variation could not be controlled, potentially leading to overestimation of genetic effects [40]. Second, the absence of genetic parameters (e.g., heritability estimates) limited the ability to assess the inheritability of the traits [41]. Furthermore, this study primarily focused on analyzing correlations between morphological and mass traits. The cross-sectional data structure and sampling from a single wild population limited our ability to rigorously investigate growth patterns in terms of allometry and isometry. Addressing this fundamental aspect of growth dynamics will require future studies incorporating ontogenetic series or multi-population comparisons.

To address these limitation, Future studies should adopt family-based designs or repeated measurements to more accurately partition genetic and environmental variance. Additionally, integrating genomic approaches such as GWAS or transcriptome analysis, or QTL mapping could facilitate the identification of genetic loci and functional genes underlying target traits, providing a foundation for marker-assisted selection [42,43]. Comprehensive sampling across diverse populations coupled with standardized cultivation experiments will be essential to systematically elucidate the allometric/isometric principles governing growth patterns.

Despite these limitations, this research presents the first comprehensive assessment of the breeding potential of morphological and mass traits in *S. bisulcatum*, identifying key target traits and their mechanistic underpinnings. The findings provide a reliable phenotypic framework and theoretical foundation for future selective breeding programs in this species.

## 5. Conclusions

In conclusion, this finding indicates that the mass traits (body mass and soft body mass) of *S. bisulcatum* possess significantly greater coefficients of variation than morphological traits, highlighting their substantial potential for selective breeding. Regression and path analyses identified ST as the key morphological trait associated with body mass, underscoring its value as a primary criterion for parental selection. For soft body mass, SW was the predominant predictor. Furthermore, SH and AH exhibited significant synergistic effects on body mass and soft body mass, respectively. Therefore, an integrated breeding strategy prioritizing these major traits while incorporating synergistic ones is recommended to maximize genetic gain. Future research should integrate environmental factors with genetic analyses to decipher the genetic architecture of these traits and their genotype-by-environment interactions, thereby providing a robust foundation for sustainable aquaculture of *S. bisulcatum*.

## Figures and Tables

**Figure 1 animals-15-03027-f001:**
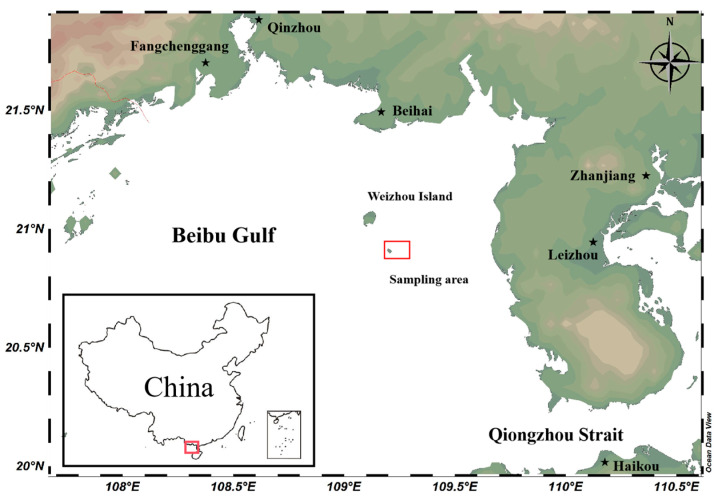
Map of the sampling area.

**Figure 2 animals-15-03027-f002:**
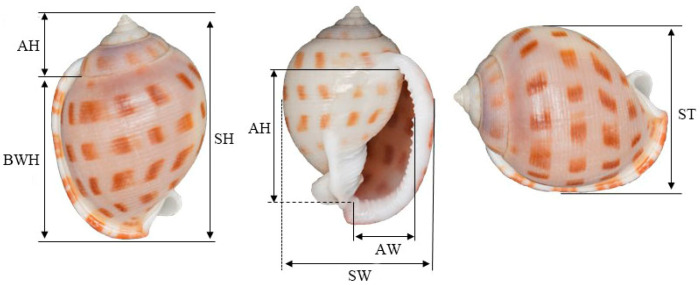
Morphological parameters of *S. bisulcatum*.

**Figure 3 animals-15-03027-f003:**
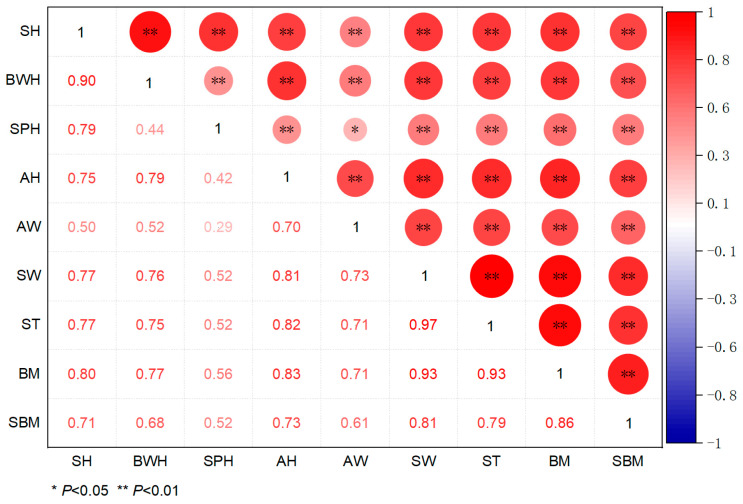
Correlation coefficients of various morphological traits.

**Figure 4 animals-15-03027-f004:**
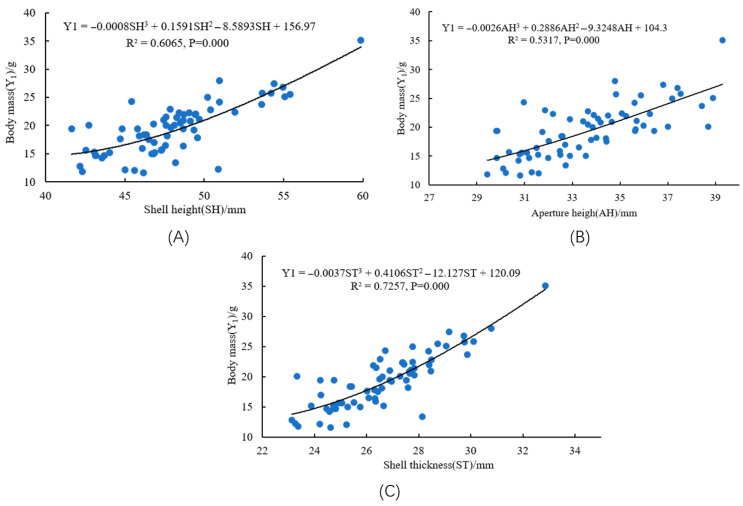
Curve fitting results of the relationships between key morphological traits and body mass of *S. bisulcatum*. (**A**) modal curve of SH versus body mass; (**B**) modal curve of AH versus body mass; (**C**) modal curve of ST versus body mass.

**Figure 5 animals-15-03027-f005:**
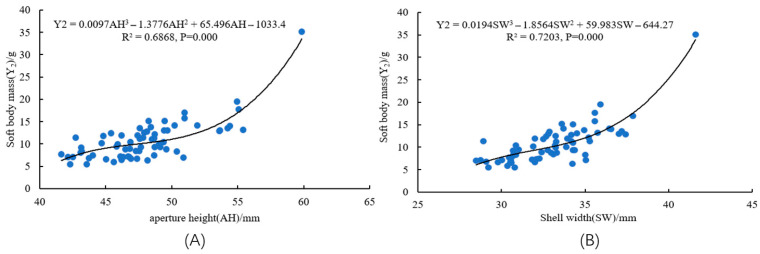
Curve fitting results of the relationships between key morphological traits and soft body mass of *S. bisulcatum*. (**A**) modal curve of AH versus oft body mass; (**B**) modal curve of SW versus soft body mass.

**Table 1 animals-15-03027-t001:** Description of the morphometric parameters and qualitative traits of *S. bisulcatum*.

No.	Parameters	Abbreviation	Description
1	Shell height	SH	The maximum linear dimension along the anteroposterior axis (from the apex to the most ventral point of the aperture).
2	Body whorl height	BWH	The linear height of the body whorl (the last and largest whorl), measured from the beginning of its suture to the ventral edge of the aperture.
3	Spire height	SPH	The height of the spire, comprising all whorls excluding the body whorl, from the apex to the suture preceding the body whorl
4	Aperture height	AH	The maximum height of the aperture opening, measured parallel to the shell’s axis from the dorsal to the ventral side.
5	Aperture width	AW	The maximum width of the aperture opening, measured perpendicular to the aperture height.
6	Shell width	SW	The maximum linear dimension of the shell perpendicular to the anteroposterior axis, indicating the degree of shell inflation.
7	Shell thickness	ST	The thickness of the shell material, typically measured at the center of the body whorl
8	Body mass	/	The total mass of the fresh organism, including the shell and all soft tissues.
9	Soft body mass	/	The mass of the soft tissues alone after removal of the shell.

**Table 2 animals-15-03027-t002:** Descriptive statistical analysis of the morphological traits and body mass of *S. bisulcatum*.

Trait	Mean	SD	Skewness	Kurtosis	CV
BW/g	19.37	4.626	0.548	0.702	23.88
SBW/g	10.74	4.408	2.716	13.095	41.04
SH/mm	47.52	3.518	0.848	1.133	7.40
BWH/mm	37.77	2.946	−0.540	2.263	7.80
SPH/mm	9.86	2.148	1.283	4.749	21.78
AH/mm	33.03	2.617	0.435	−0.454	7.92
AW/mm	22.46	1.813	0.431	0.911	8.07
SW/mm	32.84	2.395	0.688	1.107	7.29
ST/mm	26.52	1.831	0.631	0.874	6.90

**Table 3 animals-15-03027-t003:** Regression coefficient test of morphological traits on mass traits.

Dependent Variable	Model	Regression Coefficient	Standard Error	t Test Value	*p* Value
	Constant	−43.543	3.197	−13.619	0.000
Body mass	ST	1.242	0.230	5.409	0.000
	AH	0.398	0.137	2.898	0.005
	SH	0.339	0.120	2.830	0.006
	Constant	−30.859	3.585	−8.608	0.000
Soft body mass	SW	0.696	0.158	4.414	0.000
	AH	0.545	0.151	3.610	0.001

**Table 4 animals-15-03027-t004:** Analysis of variance of multiple regression equations between morphological and mass traits.

Dependent Variable	Index	Degree Freedom	Sum of Squares	Mean Square	F Value	*p* Value
	Regression	3	1002.680	334.227	128.959	0.000
Body mass	Residual	63	163.279	2.592		
	Total	66	1165.959			
	Regression	2	229.729	77.822	66.385	0.000
Soft body mass	Residual	64	3.416	3.567		
	Total	66	680.934			

**Table 5 animals-15-03027-t005:** Path analysis of various morphological traits on mass traits.

Mass Trait	Morphometric Trait	Relative Coefficient	Direct Effect	Indirect Effect				
				SH	AH	SW	ST	Sum
Body mass	SH	0.859	0.263	-	0.165	-	0.430	0.595
	AH	0.802	0.220	0.197	-	-	0.385	0.582
	ST	0.898	0.509	0.222	0.167	-	-	0.389
Soft body mass	AH	0.759	0.394	-	-	0.365	-	0.365
	SW	0.780	0.482	-	0.298	-	-	0.298

Note: “-” means no indirect effect, and no statistical significance is expressed in the interconnection of the blank.

**Table 6 animals-15-03027-t006:** Determination coefficients of morphological traits on mass traits.

Mass Trait	Morphometric Trait	SH	AH	SW	ST
Body mass	SH	0.067	0.087	-	0.226
	AH		0.048	-	0.170
	ST			-	0.259
Soft body mass	AH	-	0.154	0.286	-
	SW	-		0.232	-

## Data Availability

The original contributions presented in this study are included in the article.

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
