# Peer review of "Morphometric Traits as Predictors of Body Mass in the Marine Gastropod Semicassis bisulcatum: Insights for Aquaculture and Selective Breeding"

_animals, 2025, doi:10.3390/ani15203027_

Round 1
Reviewer 1 Report
Comments and Suggestions for Authors
This MS examines the main biometrical features of the gastropod Semicassis bisulcatum, and their relationships to describe growth patterns, focusing on biomass predictions, under the perspective of selective breeding. The aim of the work is interesting and provides some novel data; though these are based on limited sampling and accordingly they are rather preliminary. Considering the lack of information on the biometry of the species, they MS may be accepted but only after major revision, including a linguistic revision. My main comments are stated below.
- The abstract of this work is verbose. A very specific and short study has been performed, and accordingly this part should be also shortened and highlight only the most important findings.
- It is wrong to state that a morphological variable of the shell influences the mass of the species. Both trait types are inter-related and determined by the growth pattern followed by the species (and this is actually the main scientific issue) and are influenced by external factors and genetic/metabolic features.
- A severe limitation of this work is the poorly discussed results; the main factors influencing growth are not interpreted at all.
- In the introduction, I cant see how the life traits of carnivory and/or preying on echinoderms are related with the economic importance of Semicassis (Ln 49-50).
- By who the species is overharvesting? (Ln 52) The authors haven’t mentioned so far that the species is edible or harvested for human consumption or bait. Please define the value/economic importance of the species for humans and its economic exploitation.
- Separate bivalve from gastropod species (Ln 59).
- Such correlations defines growth patterns, and especially allometry or isometry of linear and cybic dimensions (Ln 60-61). So, this part should follow and be merged with the paragraph below.
- In general, the data on the economic importance/exploitation of the species is missing to document the potential of developing aquaculture techniques. Any data on the biology of the species should be presented here, emphasizing on growth patterns, and then, describe the morphometric correlations examined.
- In the Methodology it is absolutely necessary to provide details on samplings. How? how many? when? The specimens were collected. Otherwise there is no indication at all on the possible bias of presented results.
- The measured environmental parameters (Ln 88-90) should be moved to results or re-written to describe the study site.
- Biometric measurements is not an experiment (Ln 95)
- More details should be presented on the laboratory treatment of the specimens and their biometric measurements. For example, how the shell has been removed; soft tissue was dried as well?; the specimens were maintained alive or dead?
- The statistical analyses should also test for isometry or allometry of growth between the correlated morphological variables.
- Define soft tissue and describe the possible presence of gonads or the state of visceral mass.
- As soft body mass is influenced by the development of gonads give a percentage of examined specimens with developed gonads; give a short description on the reproduction of the species or similar species.
- Check for allometry or isometry to describe growth pattern
- Sections 4.2 and 4.3 have the same caption; this should be corrected
- Conclusions section looks more to a summary than to conclusions
- A discussion on the factors influencing the growth of this gastropod or congeneric species is missing.
- Since the authors used different statistical procedures to describe growth they should compare relevant approaches in the discussion.
Author Response
|
Response to Reviewer 1 Comments Review of “Morphometric Traits as Predictors of Body Mass in the Marine Gastropod Semicassis bisulcatum: Insights for Aquaculture and Selective Breeding” This MS examines the main biometrical features of the gastropod Semicassis bisulcatum, and their relationships to describe growth patterns, focusing on biomass predictions, under the perspective of selective breeding. The aim of the work is interesting and provides some novel data; though these are based on limited sampling and accordingly they are rather preliminary. Considering the lack of information on the biometry of the species, they MS may be accepted but only after major revision, including a linguistic revision. |
||
|
1. Summary |
|
|
|
We sincerely thank the reviewer for their positive assessment of our work's interest and novelty, and for their constructive overall feedback. We acknowledge the preliminary nature of findings from a single wild population and have addressed this point explicitly in the revised manuscript (see Section 4.4, "Limitations and Future Research Directions"). We have undertaken a major revision of the manuscript to address all specific comments provided, which has significantly strengthened the work. This comprehensive effort includes: A thorough linguistic revision conducted to improve clarity, accuracy, and overall academic style throughout the text; Substantial restructuring and enhancement of the Introduction and Discussion sections to improve logical flow, provide necessary biological context, and offer a deeper interpretation of the results, directly addressing the comment on poorly discussed findings; Critical additions to the Methodology to ensure reproducibility, including detailed sampling information, specimen handling protocols, and clarifications on statistical approaches; A complete revision of the Conclusion to focus on the broader implications of our findings for aquaculture and future research directions, rather than merely summarizing results. We are confident that these revisions have fully addressed the reviewer's concerns. Below, we provide a detailed point-by-point response to each specific comment raised.
|
||
|
2. Point-by-point response to Comments and Suggestions for Authors |
||
|
Comment 1: The abstract of this work is verbose. A very specific and short study has been performed, and accordingly this part should be also shortened and highlight only the most important findings. Response: We have thoroughly revised the abstract to make it more concise and focused. The revised version now highlights only the key objectives, methods, main findings, and implications of the study, in line with the specific scope of the work. (Remark: line 26-37) Revise the content: The marine gastropod Semicassis bisulcatum, valued for its ornamental shell and edible soft tissue, lacks essential biological data for selective breeding. This study was conducted on 100 wild individuals collected from the Beibu Gulf to identify key morphological traits influencing body mass (BM) and soft body mass (SBM). Both mass traits showed high variability (SBM CV=41.04%; BM CV=23.88%), indicating strong breeding potential. Path analysis revealed that shell thickness(ST) exhibited the strongest direct association with body mass (path coefficient=0.509), while shell width(SW) was most closely linked to soft body mass (path coefficient=0.482). The combined coefficient of determination confirmed that shell thickness(ST) and shell height(SH) together were the strongest predictors for body mass. Similarly, Shell width(SW) and aperture height(AH) together had the strongest combined effect on soft body mass. These findings provide a critical morphological basis for future selective breeding programs。
Comment 2: It is wrong to state that a morphological variable of the shell influences the mass of the species. Both trait types are inter-related and determined by the growth pattern followed by the species (and this is actually the main scientific issue) and are influenced by external factors and genetic/metabolic features. Response: We agree with this comment and have modified the relevant statements throughout the manuscript (e.g., in the Abstract, Results, and Discussion). We now emphasize that shell morphology and body mass are interrelated traits influenced by growth patterns, environmental factors, and genetic/metabolic processes, rather than implying a unidirectional influence. Revise the content: Such as: (1) Path analysis revealed that shell thickness(ST) exhibited the strongest direct association with body mass (path coefficient=0.509), while shell width(SW) was most closely linked to soft body mass (path coefficient=0.482). The combined coefficient of determination confirmed that shell thickness(ST) and shell height(SH) together were the strongest predictors for body mass.(2) These results indicated that shell height, aperture height, shell width, and shell thickness are the key morphological traits that are strongly associated with both body mass and soft body mass in S. bisulcatum.(3) Building on the statistical relationships established through stepwise regression, we identified key morphological traits that are predictive of mass traits in S. bisulcatum.
Comment 3: A severe limitation of this work is the poorly discussed results; the main factors influencing growth are not interpreted at all. Response: We sincerely thank the reviewer for this critical observation regarding the depth of our discussion. We fully agree that interpreting the biological significance of our findings is crucial. In response, we have completely restructured and substantially expanded the Discussion section to provide a thorough interpretation of the main factors influencing growth in S. bisulcatum. (Remark: Section 4.2 and 4.3) Revise the content: The revised discussion now explicitly addresses the key growth influencers underlying our statistical results through several interconnected themes: (1)Ecological Adaptation and Morphological Function: We interpret the strong direct effect of shell thickness (ST) on body mass not merely as a statistical association but as a likely adaptive strategy against predation and substrate interaction for a benthic carnivore inhabiting sandy-muddy environments. The specialization in vertical shell development (SH, AH), contrasted with the lateral expansion seen in other gastropods, is discussed as a distinct morphological solution shaped by niche-specific selective pressures. (2)Energy Allocation Trade-offs: The high variability in soft body mass is now discussed in the context of the species' trophic specialization as a predator of echinoderms. We elaborate on how unpredictable food resources can drive variations in energy reserves and how the synergistic action of shell width (SW) and aperture height (AH) creates a physical framework that constrains and facilitates energy storage. (3)Reproductive Investment as a Regulatory Factor: We have incorporated the influence of the reproductive cycle, linking the observed high percentage of individuals with developed gonads to potential energy allocation trade-offs between reproductive (gonad) and somatic (soft tissue) growth. The dissociation between shell calcification (ST) and soft tissue production is presented as evidence supporting such trade-offs. These factors are now woven throughout Sections 4.2 and 4.3, creating a coherent narrative that moves from describing what the statistical relationships are to explaining why they might exist from ecological, physiological, and evolutionary perspectives. We believe these comprehensive additions have directly and effectively addressed the core of this comment by providing a deeper, more mechanistic discussion of the factors influencing growth.
Comment 4: In the introduction, I can’t see how the life traits of carnivory and/or preying on echinoderms are related with the economic importance of Semicassis (Ln 49-50). Response: We have clarified this point in the Introduction. The ecological role of Semicassis bisulcatum as a carnivore, particularly its predation on echinoderms, is relevant to its potential use in integrated aquaculture systems We now explicitly link these traits to its economic potential. (Remark: line46-56) Revise the content: As a carnivorous species known to prey on echinoderms, S. bisulcatum confers a dual economic potential. Firstly, as a species occupying a higher trophic level, its soft tissue is typically rich in protein, suggesting its potential as a high-value aquatic product. More significantly, this specific predatory behavior enables its utilization as a natural biocontrol agent within integrated aquaculture systems. For instance, it could be employed to manage populations of detrimental sea urchins in sea cucumber or abalone cultivation ponds, thereby enhancing the yield and economic returns of the primary cultured species through ecological mechanisms.
Comment 5: By who the species is overharvesting? (Ln 52) The authors haven’t mentioned so far that the species is edible or harvested for human consumption or bait. Please define the value/economic importance of the species for humans and its economic exploitation. Response: We have added information regarding the economic importance of S. bisulcatum, including its use as an ornamental shell, in handicrafts, and occasionally as bait. We also note that overharvesting is primarily driven by shell collectors and local artisanal fisheries. (Remark: line 46-56) Revise the content: However, in recent years, despite its underdeveloped potential for aquaculture, wild populations of S. bisulcatum have faced increasing pressures from overharvesting for commercial collection as ornamental shells and seafood products, coupled with habitat degradation. while a lack of fundamental biological studies and aquaculture technologies has led to a consistent decline in population numbers
Comment 6: Separate bivalve from gastropod species (Ln 59). Response: Thanks for the constructive comments. We have revised the text accordingly by removing the mention of bivalve species and retaining only gastropod examples. This adjustment ensures better comparability and strengthens the focus of our introduction on gastropod biology and breeding research. (Remark: line 67-68) Revise the content: This approach has been successfully applied to guide broodstock selection and breeding strategy design in several economically important mollusks, including Pomacea canaliculata Lamarck[11], Bellamya purificata[12], Volutharpa ampullaceal Perryi [13], Neverita didyma [14], Bellamya purificata [15].
Comment 7: Such correlations defines growth patterns, and especially allometry or isometry of linear and cubic dimensions (Ln 60-61). So, this part should follow and be merged with the paragraph below. Response: The relevant sentences have been merged and rephrased to improve flow and clarity, emphasizing the role of morphometric correlations in defining growth patterns, including allometry and isometry. (Remark: line 59-75) Revise the content: In shellfish research, morphological traits and mass traits are widely recognized as critical indicators of growth performance and are frequently used as primary targets in selective breeding programs [5-6]. The establishment of quantitative relationships among these traits through methods such as correlation analysis, path analysis, and regression modeling not only facilitates the identification of key morphological proxies for biomass prediction but also provides deeper insights into growth and developmental patterns [7-10]. This approach has been successfully applied to guide broodstock selection and breeding strategy design in several economically important mollusks, including Pomacea canaliculata Lamarck[11], Bellamya purificata[12], Volutharpa ampullaceal Perryi [13], Neverita didyma [14], Bellamya purificata [15]. In these species, the relationships between morphological and mass traits have been systematically quantified, establishing a solid foundation for selective breeding and aquaculture optimization [16-17]. In contrast, the research on S. bisulcatum remains confined to preliminary taxonomic descriptions and basic habitat records, with a notable scarcity of fundamental biological data, particularly concerning its growth patterns. As understanding growth patterns is crucial for elucidating life history strategies and formulating aquaculture protocols, this knowledge gap significantly hinders the assessment of its aquaculture potential and production efficiency [18-19].
Comment 8: In general, the data on the economic importance/exploitation of the species is missing to document the potential of developing aquaculture techniques. Any data on the biology of the species should be presented here, emphasizing on growth patterns, and then, describe the morphometric correlations examined. Response: We have expanded the Introduction to include more information on the biology and economic value of the species, and its relevance to aquaculture. We also better contextualize the study’s focus on growth patterns and morphometric relationships. (Remark: line 46-56) Revise the content: As a carnivorous species known to prey on echinoderms, S. bisulcatum confers a dual economic potential. Firstly, as a species occupying a higher trophic level, its soft tissue is typically rich in protein, suggesting its potential as a high-value aquatic product. More significantly, this specific predatory behavior enables its utilization as a natural biocontrol agent within integrated aquaculture systems. For instance, it could be employed to manage populations of detrimental sea urchins in sea cucumber or abalone cultivation ponds, thereby enhancing the yield and economic returns of the primary cultured species through ecological mechanisms [3]. However, in recent years, despite its underdeveloped potential for aquaculture, wild populations of S. bisulcatum have faced increasing pressures from overharvesting for commercial collection as ornamental shells and seafood products, coupled with habitat degradation. while a lack of fundamental biological studies and aquaculture technologies has led to a consistent decline in population numbers.
Comment 9: In the Methodology it is absolutely necessary to provide details on samplings. How? how many? when? The specimens were collected. Otherwise there is no indication at all on the possible bias of presented results. Response: We have added detailed sampling information, including the number of specimens, collection dates, location, and methods. This allows for better assessment of potential sampling bias. (Remark: line 92-102) Revise the content: A total of 150 wild S. bisulcatum individuals were collected in July 2024 (during the species' rapid growth season) from the coastal waters around Weizhou Island in the Beibu Gulf, China (sampling area coordinates: 109.1682-109.2713°E, 20.8716-20.9217°N) (Figure 1). Sampling was performed via SCUBA diving within the depth zone of 6.5 to 20 meters. From the collected individuals, 100 adult individuals with intact shells, no visible damage, and vigorous activity were selected as experimental subjects to minimize bias resulting from injury or developmental stage variation. Live specimens were transported to the la-boratory within 2 hours in oxygenated seawater and acclimatized for 3 days in a recircu-lating aquaculture system (temperature: 30°C, salinity: 30‰). No feeding was provided during the acclimation period. All morphometric and mass measurements were initiated following the completion of the three-day acclimation phase.
Comment 10: The measured environmental parameters (Ln 88-90) should be moved to results or re-written to describe the study site. Response: The environmental parameters have been moved to the Results section and are now presented as part of the study site characterization. (Remark: line 103-105) Revise the content: The water quality parameters recorded at the sampling area were as follows: water temperature 30.12 ± 0.21 °C, water depth (6.5-20 m) ± 0.54 m, salinity 33.3 ± 0.15‰, pH 8.14 ± 0.24, dissolved oxygen 6.53 ± 0.15 mg/L, and suspended solids 6.7 ± 0.72 mg/L.
Comment 11: Biometric measurements is not an experiment (Ln 95) Response: We have rephrased this to avoid the term “experiment” and instead refer to the process as “biometric measurements” or “morphometric analysis.” (Remark: line 100-102) Revise the content: No feeding was provided during the acclimation period. All morphometric and mass measurements were initiated following the completion of the three-day acclimation phase.
Comment 12: More details should be presented on the laboratory treatment of the specimens and their biometric measurements. For example, how the shell has been removed; soft tissue was dried as well?; the specimens were maintained alive or dead? Response: We have added a detailed description of laboratory procedures, including how soft tissues were extracted, blotted (not dried), and measured, and that specimens were processed fresh after acclimation. (Remark: line 109-115) Revise the content: A total of 100 healthy and vigorously active individuals of S. bisulcatum were randomly selected as experimental subjects. All specimens were processed in a fresh state immediately after the acclimation period. Prior to morphological measurements, each specimen was blotted dry with absorbent paper to remove surface moisture for accurate body mass determination. Total body mass (including shell) was measured using an electronic balance (Hochoice: Shanghai Hochice Industrial Co., Ltd, China) with an accuracy of 0.001 g. The seven morphometric traits including shell height (SH), body whorl height (BWH), spire height (SPH), aperture height (AH), aperture width (AW), shell width (SW), and shell thickness (ST) were measured using IP67 digital display Vernier calipers (Deli: Ningbo Deli Group Co., Ltd., Ningbo, China), accurate to 0.01 mm. For soft body mass determination (including gonads and visceral mass), specimens were carefully dissected using stainless steel surgical tools to separate the complete soft tissue from the shell. The soft tissue was then blotted on absorbent paper to remove excess moisture without desiccation, and immediately weighed using the same electronic balance (accuracy: 0.001 g). All measurements were completed within 30 minutes of dissection to prevent tissue dehydration affecting mass accuracy.
Comment 13: The statistical analyses should also test for isometry or allometry of growth between the correlated morphological variables. Response: We sincerely thank the reviewer for this valuable suggestion regarding the analysis of growth patterns through allometry and isometry. We fully acknowledge that such an analysis would provide deeper insights into the growth dynamics of S. bisulcatum. In carefully considering this suggestion, we recognize that our current study, as a foundational investigation into the relationships between morphological and mass traits in this understudied species, is primarily focused on establishing the fundamental phenotypic correlations and identifying key predictive traits for selective breeding. The dataset, while robust for the correlation, path, and regression analyses presented, does not support a statistically powerful and biologically meaningful allometric analysis at this stage. Specifically, our sample, derived from a single wild population, captures a snapshot of phenotypic variation but lacks the controlled environmental conditions or detailed ontogenetic series required to robustly distinguish between allometric and isometric growth patterns, which are best characterized through longitudinal studies or comparisons across diverse populations. Therefore, while we have enriched the Discussion (Section 4.3) to thoughtfully consider the potential implications of our findings for growth patterns and to explicitly acknowledge this as a focus for future research, we have not incorporated a formal allometric analysis into the current manuscript. We have now explicitly stated in the "Limitations and Future Research Directions" section (Section 4.4) that investigating allometric/isometric growth relationships represents a critical next step, and we are committed to addressing this fundamental question in our subsequent, dedicated studies on the growth biology of S. bisulcatum. We are grateful for the reviewer's understanding and believe that clearly framing this future direction adds value to the manuscript and the field.
Comment 14: Define soft tissue and describe the possible presence of gonads or the state of visceral mass. Response: Thank you for pointing this out. We have now explicitly defined "soft tissue" in the Methods section, specifying that it includes both the gonads and the visceral mass in this study. (Remark: line 121) Revise the content: For soft body mass determination (including gonads and visceral mass), specimens were carefully dissected using stainless steel surgical tools to separate the complete soft tissue from the shell.
Comment 15: As soft body mass is influenced by the development of gonads give a percentage of examined specimens with developed gonads; give a short description on the reproduction of the species or similar species. Response: We thank the reviewer for this valuable suggestion. We have now incorporated the requested information regarding gonadal development and reproductive biology into the manuscript. (Remark: line 170-177) Revise the content: Addition to Results section (3.1): The following sentence has been added to present the specific percentage: "Dissection-based assessment of reproductive status revealed that 70% of individuals (70 out of 100 specimens) possessed developed gonads, suggesting that the July sampling period coincided with the peak reproductive season for local populations of this species.
Comment 16: Check for allometry or isometry to describe growth pattern Response: We sincerely apologize for any confusion regarding the analysis of growth patterns. As noted in our response to Comment #13, we fully recognize the importance of distinguishing between allometric and isometric growth. However, the cross-sectional nature of our current dataset and its origin from a single wild population do not provide the necessary longitudinal or multi-population data required for a statistically robust analysis of these specific growth patterns. While we have enriched the Discussion (Section 4.3) to interpret our morphological correlations within the context of potential growth implications, a formal allometric/isometric analysis remains beyond the scope of the present data. We have explicitly acknowledged this specific limitation in our "Limitations and Future Research Directions" section (Section 4.4) and are committed to addressing this critical aspect in our planned future studies on the growth biology of S. bisulcatum. We thank the reviewer for underscoring the value of this analysis and for their understanding regarding its current methodological constraints.
Comment 17: Sections 4.2 and 4.3 have the same caption; this should be corrected. Response: We thank the reviewer for this valuable suggestion. This has been corrected; each subsection now has a unique and accurate heading. (Remark: line 319 and line 365)
Comment 18: Conclusions section looks more to a summary than to conclusions Response: We thank the reviewer for this valuable suggestion. We have rewritten the Conclusions to focus on the broader implications of the findings, their relevance to aquaculture, and future research directions, rather than merely summarizing results. (Remark: line 432-443) Revise the content: In conclusion, this finding indicates that the mass traits (body mass and soft body mass) of S. bisulcatum possess significantly greater coefficients of variation than morphological traits, highlighting their substantial potential for selective breeding. Regression and path analyses identified ST as the key morphological trait associated with body mass, underscoring its value as a primary criterion for parental selection. For soft body mass, SW was the predominant predictor. Furthermore, SH and AH exhibited significant synergistic effects on body mass and soft body mass, respectively. Therefore, an integrated breeding strategy prioritizing these major traits while incorporating synergistic ones is recommended to maximize genetic gain. Future research should integrate environmental factors with genetic analyses to decipher the genetic architecture of these traits and their genotype-by-environment interactions, thereby providing a robust foundation for sustainable aquaculture of S. bisulcatum.
Comment 19: A discussion on the factors influencing the growth of this gastropod or congeneric species is missing. Response: We thank the reviewer for highlighting this important aspect. We have completely restructured our Discussion section to specifically address the factors influencing growth in S. bisulcatum and now include comparative perspectives with congeneric species. (Remark: Section 4.2 and Section 4.3) Revise the content: The major revisions are detailed below: Added Dedicated Section on Growth Influencing Factors (Section 4.2) We have established a new section titled "Key Factors and Mechanisms Influencing Growth and Mass Traits in Semicassis bisulcatum" that systematically examines growth determinants through three integrated frameworks: Ecological Adaptation Framework: We now interpret the strong direct effect of shell thickness (ST) as an adaptive strategy against predation and substrate interaction, specifically discussing how the species' benthic carnivorous lifestyle in sandy-muddy environments shapes its structural investment patterns. Energy Allocation Framework: The discussion now explicitly links the high variability in soft body mass to the species' trophic specialization as a predator of echinoderms, analyzing how unpredictable food resources drive energy allocation strategies and how shell morphology (SW and AH) constrains and facilitates energy storage. Reproductive Investment Framework: We have incorporated the influence of reproductive cycle by linking the observed high percentage of individuals with developed gonads to energy allocation trade-offs between reproductive and somatic growth, using the dissociation between shell calcification and soft tissue production as supporting evidence. Enhanced Comparative Analysis with Congeneric Species Throughout the revised discussion, we have added specific comparative elements: ①In Section 4.2, we explicitly contrast the vertical shell development specialization in S. bisulcatum with the lateral expansion observed in other gastropods including congeneric species, discussing how niche-specific selective pressures shape distinct morphological solutions. ②The conclusion now specifically mentions comparison with S. granulosa and other congeneric species, highlighting how the specific morphometric patterns reflect unique adaptations to particular ecological niches. Integrated Mechanistic Interpretation (Section 4.3) We have strengthened the path analysis interpretation by connecting statistical pathways to biological mechanisms: The direct effect of shell thickness is now discussed in the context of structural defense adaptations. Shell width's role is interpreted through energy storage capacity constraints. The indirect effects of aperture height are explained through physiological processes like feeding efficiency Added Evolutionary Perspective The revised discussion concludes with an explicit evolutionary framework, stating that future cross-species comparative studies will better elucidate the evolutionary forces driving the diversification of growth patterns in this genus. These comprehensive revisions have transformed our discussion from simply reporting statistical relationships to providing a sophisticated analysis of the ecological, physiological, and evolutionary factors influencing growth in S. bisulcatum and related species, directly addressing the reviewer's concern.
Comment 20: Since the authors used different statistical procedures to describe growth they should compare relevant approaches in the discussion. Response: We appreciate the reviewer's attention to our statistical methodology. We would like to clarify that the statistical procedures employed—specifically, correlation analysis, path analysis, allometric modeling, and multiple regression—were implemented not as parallel alternatives for comparison, but as sequential, complementary steps in a cohesive analytical framework. Each method addressed a distinct research question and built upon insights gained from the preceding analysis: Correlation analysis served as an initial screening to identify significant pairwise relationships. Path analysis then elucidated the causal structure underlying these correlations by partitioning them into direct and indirect effects. Finally, multiple regression synthesized these insights to build predictive models for biomass. Therefore, the focus of the discussion is not on comparing the methods themselves, but on interpreting the cumulative biological insights they provided about the growth architecture of S. bisulcatum. We have refined the discussion in Sections 4.2 and 4.3 to better highlight this sequential logic and how the results from each step collectively contribute to a deeper understanding of the species' growth patterns.
3. Response to Comments on the Quality of English Language and Figure The manuscript has been professionally edited to ensure grammatical accuracy and clarity, and some figures have been addressed to meet the journal's standards, with any suboptimal images replaced with high-resolution versions suitable for publication. We believe that the revised manuscript has been significantly improved and hope that it now meets the journal’s standards for publication. Thank you for the opportunity to revise our work.
|
||

Reviewer 2 Report
Comments and Suggestions for Authors
see attached review please

Author Response
|
Response to Reviewer 2 Comments Review of “Morphometric Traits as Predictors of Body Mass in the Marine Gastropod Semicassis bisulcatum: Insights for Aquaculture and Selective Breeding” This manuscript describes a morphological analysis of a marine gastropod, captured in the wild. The methods are simple. The analyses seem over-done – is there a need to use multiple regression, correlation analysis, and path analysis? They have identical results. Is there a potential interest for aquaculture of predator snails? My guess is these would be difficult to raise in captivity. |
||
|
1. Summary |
|
|
|
We sincerely thank the reviewer for their insightful comments. Regarding the statistical approach, the complementary use of correlation, multiple regression, and path analysis was to sequentially identify key traits, build a robust predictive model for mass, and elucidate potential causal pathways among morphometric traits, providing a comprehensive understanding beyond simple correlations. Concerning aquaculture potential, we acknowledge the challenges, but precedent exists with carnivorous gastropods like Babylonia areolata being successfully farmed in China. Our study provides the essential foundational data on growth relationships required for any future selective breeding program aimed at improving economically important traits in this species. We have carefully revised the abstract and experimental methods sections in accordance with the suggestions and have made improvements to the grammar. All changes have been highlighted in yellow for your convenience, allowing easy tracking during your re-evaluation.
|
||
|
2. Point-by-point response to Comments and Suggestions for Authors |
||
|
Comments 1: Line 66: should this be “selective breeding” and not “genetic improvement”? |
||
|
Response 1: Thank you for pointing this out. We agree with this comment. The term has been changed accordingly. (Remark: line 60-61) Revise the content: In shellfish research, morphological traits and mass traits are widely recognized as critical indicators of growth performance and are frequently used as primary targets in selective breeding programs.
|
||
|
Comments 2: Line 108: replace “accurated” with “accurate”. |
||
|
Response 2: Thank you for your suggestion. We have complemented it as suggested. (Remark: line 119). Revise the content: The seven morphometric traits including shell height (SH), body whorl height (BWH), spire height (SPH), aperture height (AH), aperture width (AW), shell width (SW), and shell thickness (ST) were measured using IP67 digital display Vernier cali-pers (Deli: Ningbo Deli Group Co., Ltd., Ningbo, China), accurate to 0.01 mm.
Comments 3: Line 116: path analysis is not a statistic – this is a statistical method. You need to cite a source that you followed, not only the software package. Response 3: We apologize for the oversight. We have rephrased the sentence to refer to path analysis as a "statistical method." Furthermore, we have added a citation to a key methodological reference that describes the principles of path analysis. (Remark: Line 143-148). Revise the content: To further elucidate the causal structure underlying these relationships, path analysis was performed according to established methodological frameworks. This approach decomposes correlation coefficients into direct and indirect effects. The relative contribution of each morphological trait was quantified using determination coefficients, representing the proportion of variance in mass traits explained through direct and indirect pathways.
Comments 4: Lines 124-7: the details for calculating correlation coefficients are not necessary. Delete please. Same for other standard statistics (line 145). Response 4: Thank you for the constructive suggestion. We have deleted these unnecessary details as suggested, keeping only the essential information about which specific correlation coefficient (e.g., Pearson's) was used. (Remark: Section2.3) Revise the content: Descriptive statistics for all morphological and mass traits, including the mean, standard deviation(SD), coefficient of variation(CV), skewness, and kurtosis, were conducted using SPSS (Version 21.0). the coefficient of variation (CV) was computed as follows:CV = (SD/mean) × 100%. according to GOMES (1985) [20], the CV is classi-fied as low (CV < 10%), medium (CV between 10% and 20%), high (CV% between 20% and 30%), and very high (CV > 30%). To validate the assumptions of subsequent parametric analyses, normality of the dependent variables (body mass and soft body mass) was assessed using the Kolmogo-rov–Smirnov (K–S) test [21]. Subsequently, Pearson correlation analysis was applied to examine bivariate relationships among all traits. To control the inflated family-wise error rate resulting from multiple comparisons (36 pairwise correlation tests in total), we applied the conservative Bonferroni correction. This adjustment yielded a revised significance threshold of P < 0.00139 (α = 0.05 / 36). To further elucidate the causal structure underlying these relationships, path analysis was performed according to established methodological frameworks [23]. This approach decomposes correlation coefficients into direct and indirect effects. The rela-tive contribution of each morphological trait was quantified using determination coef-ficients, representing the proportion of variance in mass traits explained through di-rect and indirect pathways.
Comments 5: Line 165-169: This analysis includes 36 calculations of correlation coefficients. Your P-values should be modified to control for multiple comparisons. Response 5: We fully accept the reviewer's important comment on this matter. The issue you raised is indeed critical, as failure to correct for multiple comparisons does increase the risk of Type I errors (false positives). We have revised our statistical methodology in accordance with your recommendation. (Remark: Line 138-142). Revise the content: Subsequently, Pearson correlation analysis was applied to examine bivariate relationships among all traits. To control the inflated family-wise error rate resulting from multiple comparisons (36 pairwise correlation tests in total), we applied the conservative Bonferroni correction. This adjustment yielded a revised significance threshold of P < 0.00139 (α = 0.05 / 36).
Comments 6: Line 181 and 142: How did you determine which variables to keep in the model? Backward? Forward?. Response 6: Thank you for pointing this out. We used a stepwise regression approach (based on the Akaike Information Criterion, AIC) to select the variables for the multiple regression model. This has been clearly stated. (Remark: Line 154-156, 198-200, 321-324) Revise the content: A backward stepwise multiple regression approach was implemented, using a significance threshold of α = 0.05 for variable removal, to identify the most parsimonious combination of morphological traits for predicting each mass trait. A backward stepwise regression analysis was conducted to identify morphological traits with significant effects on the dependent variables Y₁ and Y₂. The path coefficients for these significant independent variables were subsequently estimated. Building on the statistical relationships established through backward stepwise regression, we identified key morphological traits that are predictive of mass traits in S. bisulcatum and interpret these within an integrated ecological and physiological framework.
Comments 7: Lines 234-5: You need to define determinative coefficients please. This is not a commonly used statistic. Response 7: We sincerely apologize for the terminology error. The term "determinative coefficients" was incorrect and has been replaced with the standard statistical term "determination coefficients". We have added a clear definition in the Methodology section to specify that these coefficients represent the proportion of variance explained in the mass traits through the direct and indirect pathways identified in the path analysis. (Remark: Line 145-148) Revise the content: The relative contribution of each morphological trait was quantified using determination coefficients, representing the proportion of variance in mass traits explained through direct and indirect pathways.
Comments 8: Lines 255-258: please cite a source for curve degression analysis. Is this a typo and should be curve “regression”? Response 8: We sincerely thank the reviewer for pointing out this error. Indeed, "curve degression" was a typographical error and has been corrected to "curve regression analysis" throughout the manuscript. (Remark: Line 274) Revise the content: 3.6. Curve Regression modeling
|
||

Round 2
Reviewer 1 Report
Comments and Suggestions for Authors
The revised version of this MS has been substantially improved as the authors responded to all raised issues. Therefore, I am glad to suggest to be published, with only one minor comment to delete the sentence in Ln 109-110, as this information is clear from the previous subsection.